# The Interacting Multiple Model Filter and Smoother on Boxplus-Manifolds [note 1]

**DOI:** 10.3390/s21124164

**Published:** 2021-06-17

**Authors:** Tom L. Koller, Udo Frese

**Affiliations:** Multi-Sensor Interactive Systems Group, University of Bremen, 28359 Bremen, Germany; ufrese@informatik.uni-bremen.de

**Keywords:** IMM, RTS, smoothing, manifolds, hybrid estimation, orientation estimation, quaternion smoothing

## Abstract

Hybrid systems are subject to multiple dynamic models, or so-called modes. To estimate the state, the sequence of modes has to be estimated, which results in an exponential growth of possible sequences. The most prominent solution to handle this is the interacting multiple model filter, which can be extended to smoothing. In this paper, we derive a novel generalization of the interacting multiple filter and smoother to manifold state spaces, e.g., quaternions, based on the boxplus-method. As part thereof, we propose a linear approximation to the mixing of Gaussians and a Rauch–Tung–Striebel smoother for single models on boxplus-manifolds. The derivation of the smoother equations is based on a generalized definition of Gaussians on boxplus-manifolds. The three, novel algorithms are evaluated in a simulation and perform comparable to specialized solutions for quaternions. So far, the benefit of the more principled approach is the generality towards manifold state spaces. The evaluation and generic implementations are published open source.

## 1. Introduction

The interacting multiple model filter (IMM) is widely used in the field of target tracking. After its original invention [1] for radar-based aircraft tracking [2,3,4], it has been used in various applications such as attitude estimation [5], lane change prediction [6], and sensor fault detection [7].

The IMM is applied when a single dynamical model does not predict the behavior of the system accurately [8]. This is the case when the system dynamics depend on modes, i.e., discrete states that change abruptly. The original IMM runs one Kalman Filter (KF) per mode and fuses the estimates of the filters probabilistically based on the likelihood of the models. Up to now, several adaptations of the IMM have been published to use different nonlinear filters such as the Extended Kalman Filter (EKF) [8], the Unscented Kalman Filter (UKF) [4], or the Particle Filter (PF) [9].

The accuracy of IMM estimates can be improved by smoothing [10]. Smoothing improves the past estimate by propagating the information of measurements backwards [11]. Retrospectively, it uses measurements from the past and future. Similarly to the IMM filter, the IMM smoother uses a smoother for each mode and fuses their estimates probabilistically. Several mixing schemes already exist and further schemes are being developed [12,13].

Typically, nonlinear filters and smoothers are designed to operate on vector spaces ($R^{n}$). Thus, it is difficult to maintain manifold structures like the rotation quaternion in the filter. As it is often required to estimate orientations in target tracking, various extensions [14,15,16,17] have been developed to handle rotation quaternions or matrices correctly without destroying their manifold properties, e.g., unit norm or orthogonality. To our knowledge, no such extension exists for the IMM nor the IMM smoother.

For quaternions, the approach to normalize the quaternion exists [5]. However, such an approach usually degrades the performance of the estimator due to erroneous covariance matrices. Besides that approach, publications avoid using quaternions in the state by using 2D rather than 3D [6] or by using models that work in world coordinates only, as is common in aircraft tracking [2]. In [18], the so called delta-quaternion is used in the state, while the quaternion orientation is predicted outside of the IMM.

We see a gap in the literature on how to handle manifolds in the IMM smoother in a principled way. This paper closes the gap by proposing an IMM smoother that can handle manifolds properly. While we use quaternions as an accessible application example, we designed our novel algorithms to handle more general manifolds—e.g., a sphere.

The core difficulty of an IMM smoother on manifolds is the probabilistic mixing of states. In the IMM, the estimates of all filters are mixed in a weighted sum. Unfortunately, the operator + breaks the manifold structure, wherefore, a sum cannot be computed [14]. Similar problems appear for the covariance, which conceptually assumes additive noise on the state.

To overcome this problem for single-mode filtering on quaternions, the multiplicative EKF (MEKF) [15] or the error state KF (ESKF) [16] were developed. Both methods update the quaternion estimate by quaternion multiplication only, which sustains the manifold structure in contrast to addition. The boxplus-method (⊞-method) of Hertzberg et al. [14] generalizes this concept for manifolds. It only allows changes to the manifold, which do not break its structure. It gained attention in pose tracking in the last years since it is a general approach to handle manifolds in nonlinear filtering [19,20] and least squares optimization [21]. The ⊞-method encapsulates manifolds as black boxes, so that algorithms can handle them generically. Furthermore, it provides the necessary definitions to calculate a weighted sum of Gaussians over manifolds as it is required for the IMM on manifolds.

This paper is an extended version of [22], where we derived an IMM filter that uses the ⊞-method to properly handle the mixing of states and covariances in the manifold case. The method has been proven to perform a first-order correct probabilistic mixing of Gaussians. The additional contribution of this paper is the extension of the IMM filter on ⊞-manifolds to an IMM smoother. To do so, we extend the ⊞-method to Rauch–Tung–Striebel (RTS) smoothing [23].

In this work, we focus on the tangent spaces of manifolds to derive the RTS smoother equations and reinterpret the mixing of covariances with tangent space transforms. To achieve this, we propose a generalized definition of Gaussians on ⊞-manifolds, which clarifies how to transform Gaussians between different tangent spaces. The rigorous consideration of tangent space transformations results in a generic Extended Kalman smoother (EKS) on ⊞-manifolds. The combination of the novel EKS and the probabilistic mixing of Gaussians of [22] directly yields the IMM smoother on ⊞-manifolds based on the scheme of [11].

The remainder of the paper is structured as follows: The theoretic foundation of probabilistic mixing on ⊞-manifolds with a new derivation of the covariance mixing based on our generalized definition of Gaussians is shown in Section 2. The resulting IMM Filter on ⊞-manifolds is presented in Section 3. As a basis for the IMM smoother, an RTS smoother on ⊞-manifolds is derived in Section 4. The novel IMM smoother is shown in Section 5. In Section 6, we give a simulation example for the new algorithms and analyze their performance compared to state of the art approaches. Finally, we conclude and discuss future work in Section 7.

## 2. Weighted Sum of Gaussians on ⊞-Manifolds

Usually, the representation of a manifold S is overparametrized, i.e., represented with more parameters than it has degrees of freedom (DOF). The key idea of the ⊞-method is to allow changes of the manifolds only in the direction of the DOFs [14]. This direction is called the tangent space $V\subset R^{DOF}$. The direction of the DOFs depends on the specific instance of the manifold. Thus, the tangent space is always defined with respect to a reference instance. Small changes to the manifold instance can be expressed in the tangent space and are applied with the operator ⊞:S×V↦S. The ⊞-operator enforces the manifold structure.

The difference of two manifold instances is also expressed in the tangent space and can be derived by the complementary boxminus-operator ⊟:S×S↦V. The ⊟-operator calculates the geodesic between two manifold instances, i.e., the shortest path between them on the manifold (x⊞(y⊟x)=y). The quadruplet {S,⊞,⊟,V} is called a ⊞-manifold. The operators for commonly used manifolds can be found in [14].

The ⊞-method allows us to compound a state of multiple manifolds and vectors. The ⊞/⊟-operators of a compound manifold *x* simply apply the operators elementwise:(1)$$\begin{array}{cc} x & =\{x_{1},\cdots ,x_{n}\}, \end{array}$$(2)$$\begin{array}{cc} x⊞\delta & =\{x_{1}⊞\delta_{1},\cdots ,x_{n}⊞\delta_{n}\}, \end{array}$$(3)$$\begin{array}{cc} y⊟x & =\{y_{1}⊟x_{1},\cdots ,y_{n}⊟x_{n}\}. \end{array}$$

For vectors, the operators naturally reduce to +/−. Thus, the method allows to compound states of manifolds and vectors seamlessly, which is essential for target tracking applications.

One axiomatic property of ⊞-manifolds is that every instance *y* of the manifold can be represented in the tangent space of any other instance *x* by y=x⊞(y⊟x)[14]. Thus, all operations can be defined with respect to the tangent space of *x* although singularities may arise if *x* and *y* are distant. Since the tangent space is a vector space, standard definitions and operations can be used. By choosing the tangent space appropriately, troublesome singularities are avoided. Essentially, this is the engineering ”trick“ of the ⊞-method.

Hertzberg et al. already stated that it is not possible to compute an expected value or a weighted sum of manifolds with the classic definition since the +-operator is undefined [14]. Instead, they derived an implicit definition of the expected value:(4)$$\begin{array}{c} E\left(X⊟\bar{X}\right)=0, \end{array}$$
where E(·) computes the expected value, *X* is a random variable on S, and $\bar{X}$ is the expected value of X. Essentially, $X⊟\bar{X}$ expresses all manifolds elements in the tangent space of $\bar{X}$, which allows us to apply the standard computation of the expected value.

In the IMM, Gaussian distributions are mixed instead of simple instances. Thus, *X* is a mixture of Gaussians $X_{j}∼N\left({\bar{x}}_{j},P_{j}\right)$ with mean ${\bar{x}}_{j}$ and covariance $P_{j}$. For ⊞-manifolds, the Gaussian is defined as follows:(5)$$\begin{array}{c} N(\mu ,P):=\mu ⊞N(0,P), \end{array}$$
where $P\in R^{DOFxDOF}$. The key of this definition is that the covariance of the Gaussian is defined with respect to the tangent space of the mean. Hence, if two Gaussians have different means, their covariances refer to different tangent spaces.

In the manifold case, the weighted sum of the mean values of the Gaussians $\bar{x}$ is not guaranteed to be the expected value $\bar{X}$ of the complete distribution. However, we prove that it is at least a first-order correct approximation of $\bar{X}$.

**Theorem** **1.**
*Let X be the mixture of M normally distributed random variables $X_{j}∼N\left({\bar{x}}_{j},P_{j}\right)$ with probabilities $p_{j}$, where $X∼\sum_{i=1}^{M}p_{j}N\left({\bar{x}}_{j},P_{j}\right)$. Then, the weighted sum $\bar{x}$ of the expected values ${\bar{x}}_{j}$ (defined implicitly by $\sum_{i=1}^{M}p_{j}\left({\bar{x}}_{j}⊟\bar{x}\right)=0$) is a first-order correct approximation of the expected value $\bar{X}$ of all elements in X.*


**Proof.** (6)$$\begin{array}{cc} E(X⊟\bar{x}) & \overset{?}{\approx }E\left(X⊟\bar{X}\right)=0, \end{array}$$(7)$$\begin{array}{cc} E(X⊟\bar{x}) & =\int p\left(x\right)\cdot \left(x⊟\bar{x}\right)dx=\sum_{j=1}^{M}\int p\left(x_{j}\right)\cdot \left(x_{j}⊟\bar{x}\right)dx_{j}, \end{array}$$(8)$$\begin{array}{cc} \int p\left(x_{j}\right)dx_{j} & =p_{j}, \end{array}$$
where we use the conventions p(x)=p(X=x) and $p\left(x_{j}\right)=p\left(X_{j}=x_{j}\right)$. Each element $x_{j}$ can be expressed in the tangent space of the respective mean ${\bar{x}}_{j}$ using the axioms of ⊞-manifolds [14]:
(9)$$\begin{array}{cc} x_{j} & ={\bar{x}}_{j}⊞\delta_{j},\;\;\;\delta_{j}=x_{j}⊟{\bar{x}}_{j}, \end{array}$$
(10)$$\begin{array}{cc} E(X⊟\bar{x}) & =\sum_{j=1}^{M}\int p\left(x_{j}\right)\cdot \left(\left({\bar{x}}_{j}⊞\delta_{j}\right)⊟\bar{x}\right)dx_{j}. \end{array}$$We approximate the ⊞-operator with a first-order Taylor series around $\delta_{j}=\vec{0}$:
(11)$$\begin{array}{cc} E(X⊟\bar{x}) & \approx \sum_{j=1}^{M}\int p\left(x_{j}\right)\cdot \left(\left({\bar{x}}_{j}⊞\vec{0}\right)⊟\bar{x}+J_{j}\cdot \left(\delta_{j}-\vec{0}\right)\right)dx_{j}, \end{array}$$
(12)$$\begin{array}{cc} J_{j} & ={\left.\frac{\partial \left(\left({\bar{x}}_{j}⊞\delta_{j}\right)⊟\bar{x}\right)}{\partial \delta_{j}}\right|}_{\delta_{j}=\vec{0}}. \end{array}$$Using Equation (Equation 8) we can split into
(13)$$\begin{array}{c} \begin{array}{cc} E(X⊟\bar{x}) & \approx \sum_{j=1}^{M}\left(p_{j}\left({\bar{x}}_{j}⊟\bar{x}\right)+J_{j}\cdot \int p\left(x_{j}\right)\delta_{j}dx_{j}\right). \end{array} \end{array}$$The first summand is 0 by the implicit definition of $\bar{x}$.
(14)$$\begin{array}{cc} E(X⊟\bar{x}) & \approx \sum_{j=1}^{M}J_{j}\cdot \int p\left(x_{j}\right)\delta_{j}dx_{j}. \end{array}$$Using the definition of ${\bar{x}}_{j}$:
(15)$$\begin{array}{cc} E(X_{j}⊟{\bar{x}}_{j}) & =0=\int p\left(x_{j}\right)\left(x_{j}⊟{\bar{x}}_{j}\right)dx_{j}=\int p\left(x_{j}\right)\delta_{j}dx_{j}. \end{array}$$We can reduce to
(16)$$\begin{array}{cc} E(X⊟\bar{x}) & \approx 0. \end{array}$$Thus, the approximation is first-order correct. □

In the IMM, the mixed distribution is approximated with a Gaussian. Since the covariance of manifolds is expressed in the tangent space, the covariances cannot be mixed as in the original IMM. First, the Gaussians have to be expressed with respect to the same tangent space.

Naturally, Gaussians on ⊞-manifolds are defined in the tangent space of their expected value, as in (Equation 5). However, we can retrieve a more general definition of the Gaussian by defining it in the tangent space of an arbitrary reference *r*:(17)$$\begin{array}{cc} N_{r}\left(E_{r}\left(X\right),{\mathrm{Cov}}_{r}\left(X\right)\right) & =r⊞N(E_{r}\left(X\right),{\mathrm{Cov}}_{r}\left(X\right)), \end{array}$$(18)$$\begin{array}{cc} E_{r}\left(X\right)=E\left(X⊟r\right) & =\int (x⊟r)\cdot p(x)dx, \end{array}$$(19)$$\begin{array}{cc} {\mathrm{Cov}}_{r}\left(X\right)=\mathrm{Cov}\left(X⊟r\right) & =\int \left((x⊟r)-E(X⊟r)\right){\left((x⊟r)-E(X⊟r)\right)}^{T}\cdot p\left(x\right)dx. \end{array}$$

Choosing r=E(X) directly yields the definitions of [14]. The standard computation rules for the expectation and covariance do not necessarily hold for this generalization, e.g., $E_{r}\left(X\right)+E_{r}\left(Y\right)\neq E_{r}\left(X+Y\right)$, since the +-operator is undefined on the manifolds.

With the new definition, the reference can be arbitrarily chosen as in Figure 1. To change the reference of the Gaussian from $r_{1}$ to $r_{2}$, the Gaussian has to be transformed to the manifold space and afterwards, to the tangent space of $r_{2}$:(20)$$\begin{array}{cc} E_{r_{2}}\left(X\right) & =E(\left(r_{1}⊞N\left(E_{r_{1}}\left(X\right),{\mathrm{Cov}}_{r_{1}}\left(X\right)\right)\right)⊟r_{2}), \end{array}$$(21)$$\begin{array}{cc} {\mathrm{Cov}}_{r_{2}}\left(X\right) & =\mathrm{Cov}(\left(r_{1}⊞N\left(E_{r_{1}}\left(X\right),{\mathrm{Cov}}_{r_{1}}\left(X\right)\right)\right)⊟r_{2}). \end{array}$$

As usual, no general closed form solution exists, since the operators are nonlinear. A first-order Taylor approximation at $E_{r_{1}}\left(X\right)$ yields
(22)$$\begin{array}{cc} E_{r_{2}}\left(X\right) & \approx \left(r_{1}⊞E_{r_{1}}\left(X\right)\right)⊟r_{2}, \end{array}$$
(23)$$\begin{array}{cc} {\mathrm{Cov}}_{r_{2}}\left(X\right) & \approx J{\mathrm{Cov}}_{r_{1}}\left(X\right)J^{T}, \end{array}$$
(24)$$\begin{array}{cc} J & =\frac{\partial \left(\left(r_{1}⊞\left(E_{r_{1}}\left(X\right)+\delta \right)\right)⊟r_{2}\right)}{\partial \delta }. \end{array}$$

To retrieve the representation in Equation (Equation 5), the expected value is computed in the manifold space:(25)$$\begin{array}{cc} E(X) & \approx r_{2}⊞\left(\left(r_{1}⊞E_{r_{1}}\left(X\right)\right)⊟r_{2}\right)=r_{1}⊞E_{r_{1}}\left(X\right). \end{array}$$

Two major types of reference transformations occur in state estimation. The first type is the reference transform from a Gaussian expressed in the tangent space of $r_{1}$ to the tangent space of its expected value $r_{2}$. This reference transform is required to reestablish the standard form of Gaussians on ⊞-manifolds, as defined in [14]. We call this type “centered transform”, since it transforms the Gaussian to the tangent space of its center, i.e., $r_{2}=r_{1}⊞E_{r_{1}}\left(X\right)$.

The second type transforms a Gaussian that is expressed in the tangent space of its expected value, i.e., $E_{r_{1}}\left(X\right)=\vec{0}$, to another reference $r_{2}$. This transform is required to express two Gaussians in the same tangent space to allow further computations. We name it ”displaced transform“.

Using displaced transforms, all Gaussians of a mixture can be expressed in the tangent space of $\bar{x}$. Then, the standard covariance mixing can be applied, which results in
(26)$$\begin{array}{cc} P & =\sum_{j=1}^{M}p\left(X_{j}\right)\left({\left[{\bar{x}}_{j}⊟\bar{x}\right]}^{⊗}+J_{j}P_{j}J_{j}^{T}\right), \end{array}$$
where ${\left[\cdot \right]}^{⊗}$ denotes the outer product and $J_{j}$ as in (Equation 12). The same equation can be derived by linearization of the covariance equation like in our conference paper [22].

The first summand of (Equation 26) expresses the spread of the means in ⊞-terms. The second summand propagates the covariances of the Gaussians to the new mean. The structural difference to the original equation is the transformation of $P_{j}$ with the Jacobian $J_{j}$. In the vector case, the Jacobian equals identity. Thus, this formula is a generalization of the Gaussian mixing of vectors to ⊞-manifolds.

## 3. An IMM Filter on ⊞-Manifolds

The IMM runs a filter, e.g., an EKF or UKF, for each mode of the system. At every time step, it performs three steps: interaction, filtering, and combination [8]. The interaction mixes the estimates of all filters according to their mode and transition probabilities. The filtering performs the prediction and update of each filter. It calculates the new mode probabilities based on the measurements. The details of this step depend on the chosen filter type. The combination step combines all estimates according to their mode probability to create the output of the IMM, which is the expected state of the system.

We derive the IMM on ⊞-manifolds (⊞-IMM) by two changes:We use the ⊞-EKF [19,20] as the single mode filter. This adapts the filtering step to ⊞-manifolds. The measurement likelihood can be calculated using Gaussians on ⊞-manifolds.We use the probabilistic mixing of Gaussians of Section 2 in the interaction and combination step.

The implicit definition of the expected values on ⊞-manifolds does not give a direct rule to compute the value. It can be computed using the iterative algorithm ⊞-WeightedSum adapted from [14]:(27)$$\begin{array}{cc} \mathrm{Input}:\; & {\bar{X}}_{0},{\bar{x}}_{j},p\left(X_{j}\right)\;\;\forall j\in \left[1,M\right], \end{array}$$(28)$$\begin{array}{cc} {\bar{X}}_{k+1} & ={\bar{X}}_{k}⊞\sum_{j=1}^{M}p\left(X_{j}\right)\left({\bar{x}}_{j}⊟{\bar{X}}_{k}\right), \end{array}$$(29)$$\begin{array}{cc} \bar{x} & =\lim_{k\mapsto \infty }{\bar{X}}_{k}. \end{array}$$

In practice, the iteration is stopped when the change of the mean is small. The convergence behavior depends primarily on the choice of the initial guess ${\bar{X}}_{0}$. The algorithm is a generalization of [24] and identical to the computation of the mean on compact Lie groups [25].

To shorten the notation of covariance mixing, we define the function ⊞-WeightedCov: $$\begin{array}{cc} \mathrm{Input}:\; & \bar{x},P_{j},{\bar{x}}_{j},p\left(X_{j}\right)\;\;\forall j\in \left[1,M\right], \end{array}$$(30)$$\begin{array}{cc} J_{j} & ={\left.\frac{\partial \left({\bar{x}}_{j}⊞\delta ⊟\bar{x}\right)}{\partial \delta }\right|}_{\delta =\vec{0}}, \end{array}$$(31)$$\begin{array}{cc} P & =\sum_{j=1}^{M}p\left(X_{j}\right)\left({\left[{\bar{x}}_{j}⊟\bar{x}\right]}^{⊗}+J_{j}P_{j}J_{j}^{T}\right). \end{array}$$

The ⊞-IMM (See Table 1) can properly handle generic ⊞-manifolds and their covariances. It does not require any ad hoc implementation to mix the states as it only uses the ⊞/⊟-interface of the manifold.

## 4. An RTS Smoother on ⊞-Manifolds

Since there is no single model smoother for general ⊞-manifolds available, we derive an RTS Extended Kalman Smoother on ⊞-manifolds (⊞-EKS) in this section.

For filtering algorithms, it is suitable to express the covariance with respect to the expected value, but care must been taken at smoothing. The reason is that different estimates (the filtered, smoothed, and predicted estimate) participate in the smoothing equations. Thus, every covariance in the equations is defined on a different tangent space. They are incompatible unless they are transformed to the same tangent space. Therefore, we derive the smoother equations for ⊞-manifolds with special care of the used tangent spaces.

Our derivation of the ⊞-EKS is based on the discussion of Singer [26]. We start with the theorem of normal correlation for ⊞-manifolds. The theorem allows us to calculate the conditional probability p(X|Z) of two jointly Gaussian distributed random variables *X* and *Z*.

We express the distributions in the two different tangent spaces with references *r* for *X* and ζ for *Z*. This enables us to apply the standard theorem of normal correlation on the tangent spaces around *r* and ζ:(32)$$\begin{array}{cc} E_{r}\left(X|Z\right) & =E_{r}\left(X\right)+{\mathrm{Cov}}_{r\zeta }\left(X,Z\right){\mathrm{Cov}}_{\zeta }{\left(Z\right)}^{-1}\left[\left(Z⊟\zeta \right)-E_{\zeta }\left(Z\right)\right], \end{array}$$(33)$$\begin{array}{cc} {\mathrm{Cov}}_{r}\left(X|Z\right) & ={\mathrm{Cov}}_{r}\left(X\right)-{\mathrm{Cov}}_{r\zeta }\left(X,Z\right){\mathrm{Cov}}_{\zeta }{\left(Z\right)}^{-1}{\mathrm{Cov}}_{r\zeta }{\left(X,Z\right)}^{T}. \end{array}$$

A formal requirement for this is that X⊟r and Z⊟ζ are jointly Gaussian-distributed. For ⊞-manifolds, this almost only holds for linear approximation. We still use the notation = instead of ≈ to clarify where further approximations are required. All expected values and covariances are expressed in the tangent spaces of *r* and ζ, and ${\mathrm{Cov}}_{r\zeta }\left(X,Z\right)$ implicitly converts the reference tangent space. As a side remark, choosing ζ=E(Z) would result in the theorem of normal correlations that is used in current ⊞-estimators [14,20].

The theorem can be applied to any probability space, conditioned under some other variables *Y*:(34)$$\begin{array}{cc} E_{r}\left(X|Z,Y\right) & =E_{r}\left(X|Y\right)+{\mathrm{Cov}}_{r\zeta }\left(X,Z|Y\right){\mathrm{Cov}}_{\zeta }{\left(Z|Y\right)}^{-1}\left[\left(\underline{Z|Y}⊟\zeta \right)-E_{\zeta }\left(Z|Y\right)\right], \end{array}$$(35)$$\begin{array}{cc} {\mathrm{Cov}}_{r}\left(X|Z,Y\right) & ={\mathrm{Cov}}_{r}\left(X|Y\right)-{\mathrm{Cov}}_{r\zeta }\left(X,Z|Y\right){\mathrm{Cov}}_{\zeta }{\left(Z|Y\right)}^{-1}{\mathrm{Cov}}_{r\zeta }{\left(X,Z|Y\right)}^{T}. \end{array}$$

For readability, we underline conditional distributions if neither their expected value nor their covariance is meant. Following [26], the variables *X*, *Z*, and *Y* are interpreted in an unusual fashion: *X* is the state $x_{k}$ at time *k*, *Z* is the state $x_{k+1}$ at time k+1, and *Y* is the vector of all measurements $z_{1:k}$ up to time *k*.
(36)$$\begin{array}{cc} E_{r}\left(x_{k}|x_{k+1},z_{1:k}\right) & =E_{r}\left(x_{k}|z_{1:k}\right)+C_{k}\left[\left(\underline{x_{k+1}|z_{1:k}}⊟\zeta \right)-E_{\zeta }\left(x_{k+1}|z_{1:k}\right)\right], \end{array}$$
(37)$$\begin{array}{cc} {\mathrm{Cov}}_{r}\left(x_{k}|x_{k+1},z_{1:k}\right) & ={\mathrm{Cov}}_{r}\left(x_{k}|z_{1:k}\right)-C_{k}{\mathrm{Cov}}_{\zeta }\left(x_{k+1}|z_{1:k}\right)C_{k}^{T}, \end{array}$$
(38)$$\begin{array}{cc} C_{k}: & ={\mathrm{Cov}}_{r\zeta }\left(x_{k},x_{k+1}|z_{1:k}\right){\mathrm{Cov}}_{\zeta }{\left(x_{k+1}|z_{1:k}\right)}^{-1}, \end{array}$$
where $C_{k}$ is known as the smoother gain. These formulas essentially describe the fusion of the past $z_{1:k}$ with a hypothetically fixed next state $x_{k+1}$. The next state is actually not fixed but distributed, given all measurements $z_{1:N}$. Thus, we have to use the formulas in [26] to calculate the expectation and variance: (39)$$\begin{array}{cc} E_{r}\left(x_{k}|z_{1:N}\right) & =E\left(E_{r}\left(x_{k}|x_{k+1},z_{1:k}\right)|z_{1:N}\right), \end{array}$$(40)$$\begin{array}{cc} {\mathrm{Cov}}_{r}\left(x_{k}|z_{1:N}\right) & =E\left({\mathrm{Cov}}_{r}\left(x_{k}|x_{k+1},z_{1:k}\right)|z_{1:N}\right)+\mathrm{Cov}\left(E_{r}\left(x_{k}|x_{k+1},z_{1:k}\right)|z_{1:N}\right). \end{array}$$

Note that only the distribution $\underline{x_{k+1}|z_{1:k}}$ has to be conditioned on $z_{1:N}$ since all other values are known constants. The expected value and covariance of (Equation 39) can be calculated by standard rules:(41)$$\begin{array}{cc} E_{r}\left(x_{k}|z_{1:N}\right) & =E\left(E_{r}\left(x_{k}|z_{1:k}\right)+C_{k}\left[\left(\underline{x_{k+1}|z_{1:N}}⊟\zeta \right)-E_{\zeta }\left(x_{k+1}|z_{1:k}\right)\right]\right)\\ \; & =E_{r}\left(x_{k}|z_{1:k}\right)+C_{k}\left[E_{\zeta }\left(x_{k+1}|z_{1:N}\right)-E_{\zeta }\left(x_{k+1}|z_{1:k}\right)\right], \end{array}$$
(42)$$\begin{array}{cc} \mathrm{Cov}\left(E_{r}\left(x_{k}|x_{k+1},z_{1:k}\right)|z_{1:N}\right) & =\mathrm{Cov}\left(E_{r}\left(x_{k}|z_{1:k}\right)+C_{k}\left[\left(\underline{x_{k+1}|z_{1:N}}⊟\zeta \right)-E_{\zeta }\left(x_{k+1}|z_{1:k}\right)\right]\right)\\ \; & =\mathrm{Cov}\left(C_{k}\left(\underline{x_{k+1}|z_{1:N}}⊟\zeta \right)\right)\\ \; & =C_{k}\mathrm{Cov}\left(\underline{x_{k+1}|z_{1:N}}⊟\zeta \right)C_{k}^{T}\\ & =C_{k}{\mathrm{Cov}}_{\zeta }\left(x_{k+1}|z_{1:N}\right)C_{K}^{T}. \end{array}$$

Substituting all into Equation (Equation 40) yields
(43)$$\begin{array}{cc} {\mathrm{Cov}}_{r}\left(x_{k}|z_{1:N}\right)\; & ={\mathrm{Cov}}_{r}\left(x_{k}|z_{1:k}\right)+C_{k}\left[{\mathrm{Cov}}_{\zeta }\left(x_{k+1}|z_{1:N}\right)-{\mathrm{Cov}}_{\zeta }\left(x_{k+1}|z_{1:k}\right)\right]C_{k}^{T}. \end{array}$$

From here, it is required to choose suitable references *r* and ζ to be able to compute the smoothing step. Choosing the references is crucial for the mathematical complexity of the computation. Covariances in the ⊞-IMM all live in the tangent space of the expected value, i.e., the estimate. Thus, they are all in different tangent spaces and have to be transferred to the tangent spaces required by the smoothing step. The choice of the references decides how many transforms have to be calculated. We choose $r=E(x_{k}|z_{1:k})$ and $\zeta =E(x_{k+1}|z_{1:k})$. This avoids transforming the covariances ${\mathrm{Cov}}_{r}\left(x_{k}|z_{1:k}\right)$ and ${\mathrm{Cov}}_{\zeta }\left(x_{k+1}|z_{1:k}\right)$ as they are already available in these tangent spaces. It also reduces the expected value to
(44)$$\begin{array}{c} \begin{array}{c} E_{r}\left(x_{k}|z_{1:N}\right)=E\left(x_{k}|z_{1:k}\right)⊟E\left(x_{k}|z_{1:k}\right)+C_{k}\left[E\left(\underline{x_{k+1}|z_{1:N}}⊟E\left(x_{k+1}|z_{1:k}\right)\right)-E\left(\underline{x_{k+1}|z_{1:k}}⊟E\left(x_{k+1}|z_{1:k}\right)\right)\right]\\ =C_{k}E\left(\underline{x_{k+1}|z_{1:N}}⊟E\left(x_{k+1}|z_{1:k}\right)\right). \end{array} \end{array}$$

By linear approximation, we retrieve
(45)$$\begin{array}{cc} E_{r}\left(x_{k}|z_{1:N}\right) & \approx C_{k}\left[E\left(x_{k+1}|z_{1:N}\right)⊟E\left(x_{k+1}|z_{1:k}\right)\right]. \end{array}$$

The value of ${\mathrm{Cov}}_{\zeta }\left(x_{k+1}|z_{1:N}\right)$ can be retrieved by a displaced transform. Finally, we give a formula for ${\mathrm{Cov}}_{r\zeta }\left(x_{k},x_{k+1}|z_{1:k}\right)$ by approximation of
(46)$$\begin{array}{cc} {\mathrm{Cov}}_{r\zeta }\left(x_{k},x_{k+1}|z_{1:k}\right) & =\mathrm{Cov}(\underline{x_{k}|z_{1:k}}⊟E\left(x_{k}|z_{1:k}\right),g\left(\underline{x_{k}|z_{1:k}}\right)⊟E\left(x_{k+1}|z_{1:k}\right)), \end{array}$$
where *g* is the dynamic model function. Approximating the second argument yields
(47)$$\begin{array}{c} \begin{array}{c} {\mathrm{Cov}}_{r\zeta }\left(x_{k},x_{k+1}|z_{1:k}\right)\approx \mathrm{Cov}\left(\underline{x_{k}|z_{1:k}}⊟E\left(x_{k}|z_{1:k}\right),g\left(E\left(x_{k}|z_{1:k}\right)\right)⊟E\left(x_{k+1}|z_{1:k}\right)+F_{k}\left[\underline{x_{k}|z_{1:k}}⊟E\left(x_{k}|z_{1:k}\right)\right]\right)\\ ={\mathrm{Cov}}_{r}\left(x_{k}|z_{1:k}\right)F_{k}^{T} \end{array} \end{array}$$
(48)$$\begin{array}{cc} F_{k} & ={\left.\frac{\partial \left(g\left(E\left(x_{k}|z_{1:k}\right)⊞\delta \right)⊟E\left(x_{k+1}|z_{1:k}\right)\right)}{\partial \delta }\right|}_{\delta =\vec{0}}, \end{array}$$
where $F_{k}$ is commonly known as the state transition matrix.

With the chosen references, the complete smoothing step is (using more common symbols)
(49)$$\begin{array}{cc} {\hat{x}}_{k|N} & ={\hat{x}}_{k|k}⊞C_{k}\left[{\hat{x}}_{k+1|N}⊟{\hat{x}}_{k+1|k}\right], \end{array}$$
(50)$$\begin{array}{cc} P_{k|N} & =J_{k}\left(P_{k|k}+C_{k}\left[B_{k}P_{k+1|N}B_{k}^{T}-P_{k+1|k}\right]C_{k}^{T}\right)J_{k}^{T}, \end{array}$$
(51)$$\begin{array}{cc} C_{k} & =P_{k|k}F_{k}^{T}P_{k+1|k}^{-1}\;\;, \end{array}$$
(52)$$\begin{array}{cc} F_{k} & ={\left.\frac{\partial \left(g\left({\hat{x}}_{k|k}⊞\delta \right)⊟{\hat{x}}_{k+1|k}\right)}{\partial \delta }\right|}_{\delta =\vec{0}}, \end{array}$$
(53)$$\begin{array}{cc} B_{k} & ={\left.\frac{\partial \left(\left({\hat{x}}_{k+1|N}⊞\delta \right)⊟{\hat{x}}_{k+1|k}\right)}{\partial \delta }\right|}_{\delta =\vec{0}}, \end{array}$$
(54)$$\begin{array}{cc} J_{k} & ={\left.\frac{\partial \left(\left({\hat{x}}_{k|k}⊞\left(C_{k}\left[{\hat{x}}_{k+1|N}⊟{\hat{x}}_{k+1|k}\right]+\delta \right)\right)⊟{\hat{x}}_{k|N}\right)}{\partial \delta }\right|}_{\delta =\vec{0}}. \end{array}$$

These formulas are structurally similar to the standard RTS formulas [23]:(55)$$\begin{array}{cc} {\hat{x}}_{k|N} & ={\hat{x}}_{k|k}+C_{k}\left[{\hat{x}}_{k+1|N}-{\hat{x}}_{k+1|k}\right], \end{array}$$(56)$$\begin{array}{cc} P_{k|N} & =P_{k|k}+C_{k}\left[P_{k+1|N}-P_{k+1|k}\right]C_{k}^{T}, \end{array}$$(57)$$\begin{array}{cc} C_{k} & =P_{k|k}F_{k}^{T}P_{k+1|k}^{-1}\;\;, \end{array}$$(58)$$\begin{array}{cc} F_{k} & ={\left.\frac{\partial g(x)}{\partial x}\right|}_{x={\hat{x}}_{k|k}\;.} \end{array}$$

In the vector case, the ⊞/⊟ operators reduce to +/− and the computation of the expected value agrees completely. In contrast, the covariance computation shows important differences — the transforms with the matrices $J_{k}$ and $B_{k}$. $B_{k}$ transforms the covariance of the smoothed future state to the tangent space of the filtered predicted state. Thus, the matrices $B_{k}P_{k+1|N}B_{k}^{T}$ and $P_{k+1|k}$ are on the same tangent space, which allows them to be subtracted. $J_{k}$ conducts a centered transform to reestablish the standard Gaussian form. Nevertheless, the covariance reduces to the standard RTS formulas in the vector case as well, since $B_{k}$ and $J_{k}$ evaluate to identity. Therefore, the derived smoothing formulas can be seen as a generalization of the standard RTS formulas.

Note that the choice of the references *r* and ζ is crucial for the structure of the final formulas. Other references may alter the performance since other linearization points are used. Regardless of the choice of the references, the formulas reduce to the standard RTS formulas in the vector case, because all transformation matrices reduce to identity and the reference ζ cancels out at the computation of the expected value. Thus, countless possibilities exist to generalize the RTS formulas on ⊞-manifolds.

The presented ⊞-EKS differs from existing RTS smoothers on quaternions [17]. Since smoothing on a quaternion state is a special case of our general ⊞-EKS, it should coincide. The smoothed covariance is calculated differently from the existing smoothers as they use the vector Equation (Equation 56). The existing RTS smoothers ignore that the covariances are defined on different tangent spaces. In practice, the formulas are almost identical for quaternions if the estimates are close to each other. In that case, the ⊞/⊟-operators are almost linear; wherefore, $B_{k}$ and $J_{k}$ evaluate close to identity.

## 5. IMM RTS Smoother

Since the IMM performs an arbitrary (but well-chosen) approximation of the hybrid estimation problem, there is no single answer on how to perform the smoothing. Several smoothing schemes exist in the literature [11,12,13]. We choose the scheme of [11] as it requires only one smoothing step per mode and is straightforward to adapt to ⊞-manifolds with our results from Section 2 and Section 4. [12] pointed out that the chosen scheme uses strong approximations. However, the approach of [12] requires operations that are currently not possible on general ⊞-manifolds.

The IMM RTS smoother in [11] can be adapted to ⊞-manifolds in a similar manner as the IMM Filter. The mixture of Gaussian approximations are exchanged with the ⊞-WeightedSum and ⊞-WeightedCov algorithms. The mode-matched smoothing is exchanged with the ⊞-EKS from Section 4. This results in the ⊞-RTSIMM smoother (⊞-RTSIMMS), as shown in Table 2.

The ⊞-RTSIMMS generically operates on all differentiable ⊞-manifolds without destroying the manifold properties of the state. To our knowledge, this is the first IMM smoother that can operate properly on quaternions and other manifolds.

## 6. Example Application and Performance Discussion

We test the new algorithms in a simulated environment to provide first insights into their performance. We choose the following setup inspired by classic radar tracking: A drone flies across known terrain. It has a stereo-camera facing downwards. With the camera, it detects known landmarks in the terrain. The task is to track the position of the drone. Since the camera is mounted on the drone, its measurements are in body coordinates. Hence, it is required to estimate the orientation of the drone to make use of the measurements. The simulated drone flies a trajectory of two straight lines and two $180^{\circ }$ curves over four visible landmarks.

The drone has two different flight modes. In the first mode, it flies straight with a constant velocity. In the second mode, it flies a curve with a constant angular rate.

We model the dynamics of the drone with the state *x*:(59)$$x={\left(q_{b}^{w}\;\;{\vec{p}}_{w}\;\;{\vec{v}}_{w}\;\;{\vec{\omega }}_{w}\right)}^{T},$$
where $q_{b}^{w}$ is the rotation quaternion that rotates a world frame vector to body frame, ${\vec{p}}_{w}$ is the position in world frame, ${\vec{v}}_{w}$ is the velocity in world frame, and ${\vec{\omega }}_{w}$ is the angular rate in world frame. The straight dynamic is modeled as follows:(60)$$g_{s}\left(x,\epsilon_{s}\right)=\left(\begin{array}{c} q_{b}^{w}\\ {\vec{p}}_{w}+\left({\vec{v}}_{w}+\epsilon_{s}\right)\Delta t\\ {\vec{v}}_{w}+\epsilon_{s}\\ {\vec{\omega }}_{w} \end{array}\right),$$
where Δt=0.05 s is the time difference between time *k* and k+1. The constant turn dynamic is modeled as in [2] with a change for the orientation and angular rate:(61)$$g_{c}\left(x,\epsilon_{c}\right)=\left(\begin{array}{c} q_{b}^{w}\exp{\left(\frac{\Delta t}{2}{\vec{\omega }}_{w}\right)}^{-1}\\ {\vec{p}}_{w}+\left(\Delta tI_{3\times 3}+B\right){\vec{v}}_{w}\\ \left(I_{3\times 3}+A\right){\vec{v}}_{w}\\ {\vec{\omega }}_{w}+\epsilon_{c} \end{array}\right),$$
where exp(·) forms a quaternion from the given Euler-angle-axis [14] and $A,B\in R^{3\times 3}$, as given in [2].

For simplification, we assume that the camera measures the position of the landmark in body coordinates instead of pixel coordinates. The measurement model is
(62)$$h\left(x\right)=q_{b}^{w}\cdot {\vec{p}}_{w}\cdot {\left(q_{b}^{w}\right)}^{-1}$$

The covariance matrices can be found in Appendix A.

### 6.1. Results

The performance of seven algorithms is evaluated in the simulation:⊞-EKF: The EKF on ⊞-manifolds, as presented in [20], on the constant turn model (Equation 61).⊞**-EKS:** The RTS EKS on ⊞-manifolds, as derived in Section 4, on the constant turn model (Equation 61).(M)-EKS: The same as the ⊞-EKS, but it uses the simplified covariance smoothing (Equation 56). This algorithm uses the ⊞-EKF for filtering but its smoothing step is identical to the MEKS or the left invariant EKS on quaternions [17].⊞**-IMM:** The IMM on ⊞-manifolds, as derived in Section 3.N-IMM: An IMM that handles the manifold state such as a vector state for mixing purposes. We call this approach naive-mixing: The quaternions are summed up in parameter space and normalized afterwards (as in [5]). It uses the standard IMM equation to sum up the covariances. The mode filters of the N-IMM are ⊞-EKFs, since we want to specifically compare the ⊞-mixing from Section 2 against naive-mixing.⊞**-RTSIMMS:** The IMM RTS smoother on ⊞-manifolds, as derived in Section 5.NM-RTSIMMS: An IMM smoother that handles mixing such as the N-IMM and uses the (M)-EKS for mode-matched smoothing. This IMM basically combines the approaches in [5] and [17].

The root mean squared error (RMSE) of the position and orientation are used as performance metrics. The ⊟-operator is used to calculate the quaternion estimate error (difference as Euler-Angle-Axis). In addition, the consistency measure of [27] is computed; it states that the estimate is consistent if it resembles the true probability distribution. Hence, the expectation bias $E(\hat{x}⊟x)$ has to be 0 and the expectation of the squared Mahalonobis distances $E\left(\left|\right|\hat{x}⊟x{\left|\right|}_{P}^{2}\right)$ has to equal the DOF of the state. Both values are computed on the position and orientation, wherefore DOF=6. The averaged metrics of 100 Monte Carlo runs are shown in Figure 2.

Both IMM smoother variants caused numerical issues at the computation of the mode-matched smoothed covariances. Apparently, the subtraction of covariances results in indefinite matrices. Since the issue arises in both IMM smoothers, it does not appear to be specific for the changed covariance formula (Equation 50). To ensure positive definite covariance matrices, the eigenvalues of the smoothed matrices are set to a minimal threshold ϵ>0. All metrics of the ⊞-RTSIMMS and NM-RTSIMMS are affected by this regularization.

As expected, the RMSEs and biases of the smoother variants are lower than their filter counterparts. Remarkably, the single model algorithms perform better than the multiple model algorithms in some of the metrics. However, the IMM variants deliver overall more consistent estimates. In the remainder of our discussion, we focus on the comparison between our three, novel algorithms and their respective state-of-the-art solutions. The ⊞-EKF metrics are only used to explain different behavior between the single- and multimodel smoothers.

The ⊞-EKS and (M)-EKS perform equally on the three metrics that concern only the state, since they only differ in the covariance computation. The (M)-EKS is closer to the optimal value $E\left(\left|\right|\hat{x}⊟x{\left|\right|}_{P}^{2}\right)=6$ than the ⊞-EKS, which implies a slightly better consistency of the covariances. In contrast, the different covariance computation has an effect on the state for the IMM smoothers, as it influences the model probabilities.

The ⊞-IMM and N-IMM perform identically at the leading digits. Hence, the ⊞-mixing does not change the computation in the presented example. The IMM smoothers perform almost identically as well.

This results are unsatisfying since the theoretically justified ⊞-mixing and smoothing should yield better accuracy and consistency. Therefore, we analyze the effects of the ⊞-mixing and the changed covariance smoothing equation.

### 6.2. Discussion: ⊞-Mixing of Gaussians

The error of the naive-mixing is negligible in the presented example. To show this, we try to quantify the error induced by naive-mixing. We calculate the weighted mean of two quaternion Gaussian distributions $q_{1},q_{2}$ for different angular differences and different probabilities (see Figure 3a).

The difference is in the range of $10^{-4}$ rad for angular differences below 0.35rad (ca. $20^{\circ }$), for all probabilities. Since the IMM usually operates at small differences between the models, the error of the naive-mixing is negligible for the mean.

Similarly, the effect of the ⊞-mixing on the covariance is small (see Figure 3b). Hence, the two mixing methods differ only for high differences of the mixed quaternions. In the presented simulation example, the angular differences are small, wherefore the mixing methods have equal results.

In general, it is unlikely that the quaternion estimates of the IMM filter differ greatly. The mixing is always performed after the update step. Hence, even big differences of the dynamic models are compensated by the update. Likewise, in the IMM smoother, the mixing is performed on the updated or smoothed estimates.

The ⊞-mixing may perform better for higher differences. To analyze this, ⊞- and naive-mixing are compared to an optimal solution (see Figure 4). Since no closed form solution to mix the Gaussians exists, the optimal solution is obtained numerically. Quaternions are sampled uniformly from the Gaussians to approximate the complete distribution.

The mean and covariance error of the ⊞-mixing increase with the distance between the Gaussian means. The approaches are almost equal for small differences between the quaternions. At higher differences, ⊞- outperforms naive-mixing. Thus, ⊞-mixing is technically an improvement but its advantage is negligible due to the small differences in IMM applications.

### 6.3. Discussion: ⊞-RTS Smoothing

The ⊞-EKS is slightly less consistent than the (M)-EKS. The difference between the ⊞-EKS and (M)-EKS is the reference transformation of the covariances. Therefore, we analyze the performance of the linear approximation of the reference transform.

We compare the conducted linear approximation of the reference transform with a numerically computed optimum, where we distinguish between displaced and centered transforms (see Figure 5). For further comparison, we add the error of no transformation, which shows the behavior of the (M)-EKS. The covariance of the Gaussians has been set to $\Sigma =0.1I_{3x3}$.

The error graphs of the centered and displaced transform clearly differ, which justifies the distinction of the two transform types. The error of the displaced transform (see Figure 5b) is, for high differences of the references, several orders of magnitudes higher than the base covariance. The reason is that wrap-around issues arise, since we approach the classic singularity of quaternions close to angles of π. Note that closeness depends on the chosen covariance. With a higher covariance, the wrap-around issues occur earlier. Since the linear transformation evaluates a single point, the wrap-around is not incorporated. However, such extreme differences of the references do not usually occur during estimation.

The comparison shows that the linear approximation has a smaller error than no transformation. Thus, regarding a single smoothing step, the ⊞-EKS is more precise than the (M)-EKS. Apparently, the improvement of a single step does not improve the whole interval smoothing. To investigate the accumulated effect of the transforms, we plot the ratio of the covariance determinants before and after the reference transform (see Figure 6). In the quaternion case, the displaced transform monotonically increases the covariance and the centered transform reduces it.

In the ⊞-EKS formula, the centered transform is dominant as it encloses the whole Equation (Equation 50). Thus, the ⊞-EKS reduces the covariance marginally in comparison with the (M)-EKS. Since the covariance is already too small in the simulation, the ⊞-EKS is slightly less consistent. Otherwise, if the covariances are too high, this effect would yield more consistent results. Thus, this is neither a clear performance degradation nor an improvement.

Noticeably, the IMM smoother variants perform almost identically. Since the IMM has an overall smaller bias than the EKF, the differences between the filtered and smoothed estimates are smaller. Thus, the reduction of the covariance matrices becomes negligible. This result coincides with the analysis of Section 6.2, since the ⊞-mixing of covariances is founded on displaced transforms.

The hypothesis that the bias has a crucial influence can be further substantiated by adapting the noise of the constant turn model to reduce the bias. This is possible since the low inconsistent noise biases the estimators towards the dynamic model instead of the measurements. Thus, the bias can be drastically reduced and the difference in consistency vanishes using an additive noise on the position of $\Sigma =10\Delta_{t}I_{3x3}$ (see Figure 7).

### 6.4. Discussion: Wrap-Up

Overall, using ⊞-mixing with a first-order approximation does not give a performance boost for the IMM on quaternions. Instead, its strength is to enable a generic IMM on differentiable ⊞-manifolds. The method encapsulates the manifold properties of the state so it can be treated as a black box. Therefore, the IMM can be implemented independently of the used state representation. It does not require any ad hoc solutions to mix the states.

It has been shown that, in the quaternion case, the reference transforms are negligible if the references are close to each other. The transforms can be dropped to reduce computational and algorithmic complexity. Thus, we recommend to use the approach of [17] for quaternion smoothing. For other manifolds, the reference transform may be relevant to achieve consistent results.

Open source C++ implementations of the generic ⊞-EKS (https://github.com/TomLKoller/Manifold-RTS-Smoother, accessed on 16 June 2021) and the ⊞-IMM and ⊞-RTSIMMS (https://github.com/TomLKoller/Boxplus-IMM, accessed on 16 June 2021) are provided. The implementations use automatic differentiation [28] to calculate all required Jacobians for state mixing and for the internal ⊞-EKF [29]. They use a generic state representation to adapt to other system models. Therefore, they can be used without taking care of the heavy math in this paper. The ⊞-IMM repository also contains the presented simulation example.

## 7. Conclusions

Three, novel estimation algorithms based on the ⊞-method have been presented in this work. All algorithms properly handle the manifold structure of ⊞-manifolds such as quaternions or rotation matrices. Hence, they do not need ad hoc normalization procedures to preserve manifold structures.

The ⊞-IMM, a first-order correct IMM on ⊞-manifolds, has been derived. We described methods to calculate the weighted mean and covariance of Gaussian mixtures on ⊞-manifolds and provided necessary proofs. With these, the ⊞-method is applied to the IMM.

By following the RTS derivation of [26], the new ⊞-EKS has been derived. The ⊞-EKS uses a different smoothing update of the covariance than published smoothers for quaternions, but the expected value is calculated identically.

By combining the ⊞-EKS and the ⊞-IMM, a novel IMM smoother on ⊞-manifolds, ⊞-RTSIMMS, has been derived. To our knowledge, the ⊞-RTSIMMS is the first IMM that enables smoothing on rotation quaternions and other ⊞-manifolds in hybrid estimation.

All algorithms are evaluated on a simulated aircraft tracking example. The evaluation shows that the estimation accuracies are not improved compared to state-of-the-art smoothing methods on quaternions and ad hoc normalization procedures. For other manifolds, a practical advantage may be achieved. The presented algorithms still have high theoretical value as they can be derived from the basic definitions of the expected value and the covariance on ⊞-manifolds. Thus, they are justified, generic algorithms.

This paper extended the family of ⊞-algorithms to hybrid estimation and smoothing. The practical strength of the presented algorithms is their generality. The ⊞-method encapsulates the manifold properties. No further ad hoc implementations are required to perform mixing and smoothing, regardless of the state. Therefore, the algorithms enable the implementation of a generic library that can handle ⊞-manifold states in hybrid estimation and smoothing. A first prototype is published alongside this paper.

The presented methods are only first-order correct, which causes linearization errors. Thus, one may develop higher-order methods or use the unscented transform as approximation, which may improve the consistency of the ⊞-EKS. Presumably, the unscented transform would not significantly reduce the error, as the error compared to the numerical solutions was quite low for small distances anyway. Instead, it should be investigated whether the indefiniteness of the covariance arises from numerical issues or systematic problems in the formulas, e.g., the approximation pointed out by [12].

## Figures and Tables

**Figure 1 sensors-21-04164-f001:**
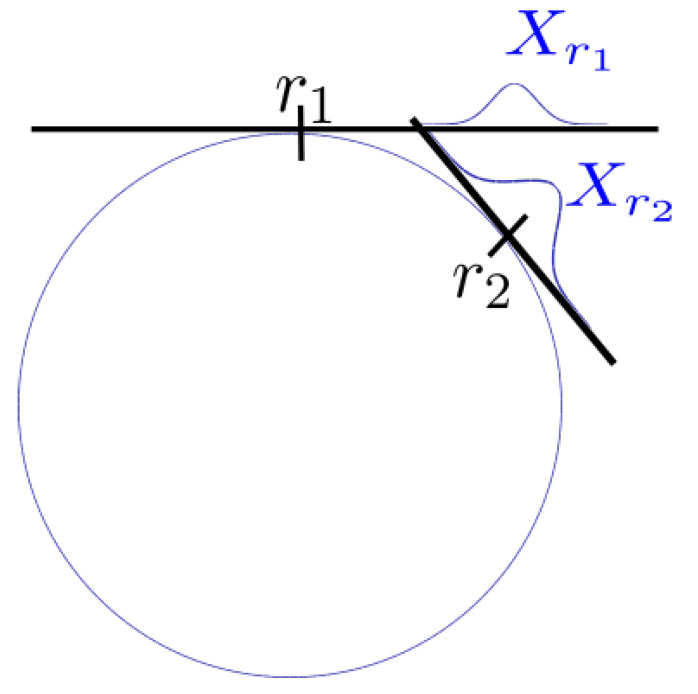
The Gaussian distribution *X* approximated in two tangent spaces on a unit circle manifold.

**Figure 2 sensors-21-04164-f002:**
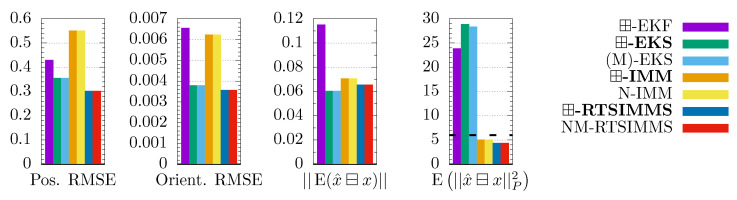
RMSE and consistency comparison for aircraft tracking in 100 Monte Carlo runs. The optimal value is 0 for $\left||E(\hat{x}⊟x)|\right|$ and 6 for $E\left(\left|\right|\hat{x}⊟x{\left|\right|}_{P}^{2}\right)$ (dashed line).

**Figure 3 sensors-21-04164-f003:**
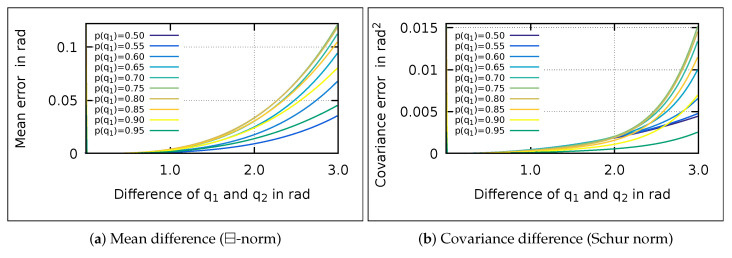
The mean and covariance difference between ⊞- and naive-mixing over the angular differences of $q_{1}$ and $q_{2}$. Reprinted with permission from ref. [22]. Copyright 2020 IEEE.

**Figure 4 sensors-21-04164-f004:**
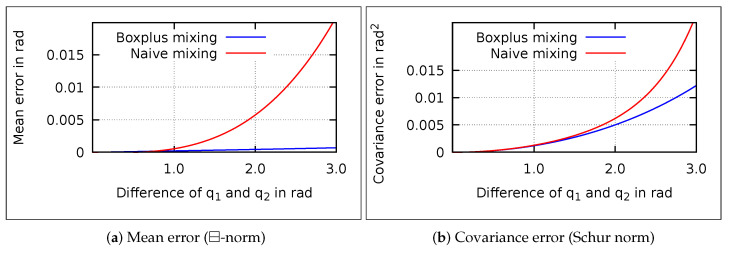
Mean and covariance error of ⊞- and naive-mixing compared to optimal mixing over the angular differences of $q_{1}$ and $q_{2}$. Reprinted with permission from ref. [22]. Copyright 2020 IEEE.

**Figure 5 sensors-21-04164-f005:**
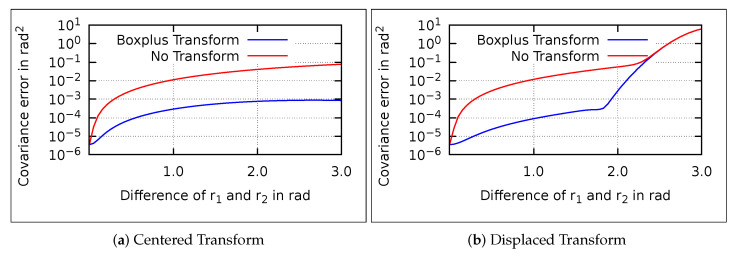
The covariance error (Schur norm) in logarithmic scale of the ⊞-transform and no transform compared to numerical optimal transformation over the angular differences of $r_{1}$ and $r_{2}$.

**Figure 6 sensors-21-04164-f006:**
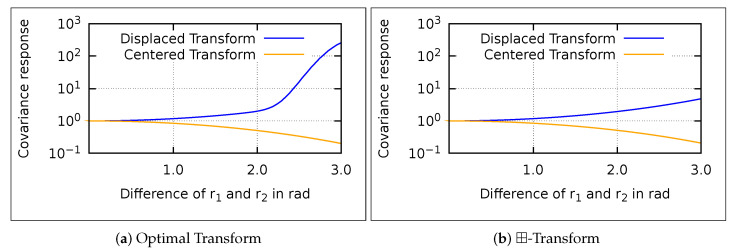
The covariance response in logarithmic scale over the angular differences of $r_{1}$ and $r_{2}$. The covariance response is the ratio of the transformed and base covariance determinants.

**Figure 7 sensors-21-04164-f007:**
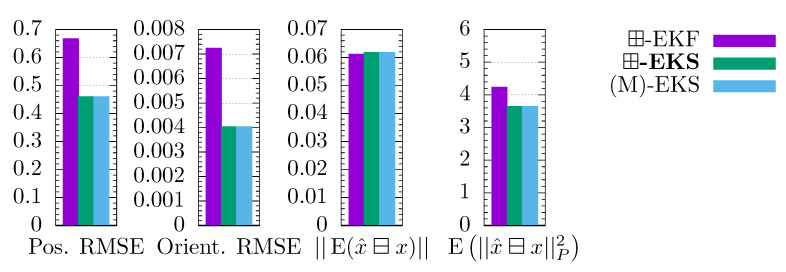
RMSE and consistency comparison of single-model estimators with additional noise on the position.

**Table 1 sensors-21-04164-t001:** Original IMM  [8] vs. ⊞-IMM. Dashed boxes belong to the standard IMM only and full boxes to the ⊞-IMM. $z_{k}$ may be a ⊞-manifold D. Modified from [22].

**State, input, process and measurement models**	$\begin{array}{cc} & x_{k}^{j}\in S,\;P_{k}^{j}\in R^{DOF\times DOF},\;u_{k}\in R^{\nu },\;z_{k}\in D\\ & g_{j}:S\times R^{\nu }\times R^{n}\mapsto S,\;x_{k+1}^{j}=g_{j}\left(x_{k}^{j},u_{k},\epsilon_{k}^{j}\right)\\ & \epsilon_{k}^{j}=N_{n}\left(0,Q_{k}^{j}\right),\;Q_{k}^{j}\in R^{n\times n}\\ & h:S\mapsto D,\;z_{k}=h\left(x_{k}\right){⊞}_{Z}N_{d}\left(0,R_{k}\right),\;R_{k}\in R^{d\times d} \end{array}$
**Initialization** ∀j∈[1,M]	$\;{\hat{x}}_{0|0}^{j}=x_{0},\;P_{0|0}^{j}=P_{0},\;\mu_{0|0}^{j}=\mu_{0}^{j}$
**Interaction** ∀i,j∈[1,M]	$c_{k}^{j}=\sum_{i=1}^{M}p_{ij}\mu_{k|k}^{i}$ $\mu_{k|k}^{ij}=\frac{p_{ij}\mu_{k|k}^{i}}{c_{k}^{j}},\;p_{ij}\;\mathrm{are}\;\mathrm{transition}\;\mathrm{probabilities}.$ $\begin{array}{c} {\hat{x}}_{k|k}^{0j}=\sum_{i=1}^{M}{\hat{x}}_{k|k}^{i}\mu_{k|k}^{ij} \end{array}$ x^k|k0j=⊞-WeightedSumi∈[1,M](x^k|kj,x^k|ki,μk|kij) $\begin{array}{c} P_{k|k}^{0j}=\sum_{i=1}^{M}\mu_{k|k}^{ij}\left(P_{k|k}^{i}+{\left[{\hat{x}}_{k|k}^{i}-{\hat{x}}_{k|k}^{0j}\right]}^{⊗}\right) \end{array}$ Pk|k0j=⊞-WeightedCovi∈[1,M]x^k|k0j,Pk|ki,x^k|ki,μk|kij
**Filtering: Prediction (**⊞**-EKF)**∀j∈[1,M]	$\begin{array}{cc} & {\hat{x}}_{k+1|k}^{j}=g_{j}\left({\hat{x}}_{k|k}^{0j},u_{k},\vec{0}\right)\\ & F_{k}^{j}={\left.\frac{\partial \left(g_{j}\left({\hat{x}}_{k|k}^{0j}⊞\delta ,u_{k},\vec{0}\right)⊟{\hat{x}}_{k+1|k}^{j}\right)}{\partial \delta }\right|}_{\delta =\vec{0}}\\ & U_{k}^{j}={\left.\frac{\partial \left(g_{j}\left({\hat{x}}_{k|k}^{0j},u_{k},\epsilon \right)⊟{\hat{x}}_{k+1|k}^{j}\right)}{\partial \epsilon }\right|}_{\epsilon =\vec{0}}\\ & P_{k+1|k}^{j}=F_{k}^{j}P_{k|k}^{0j}{\left(F_{k}^{j}\right)}^{T}+U_{k}^{j}Q_{k}^{j}{\left(U_{k}^{j}\right)}^{T} \end{array}$
**Filtering: Update (**⊞**-EKF)**∀j∈[1,M]	$\begin{array}{c} H_{k}^{j}={\left.\frac{\partial \left(h\left({\hat{x}}_{k+1|k}^{j}⊞\delta \right)⊟h\left({\hat{x}}_{k+1|k}^{j}\right)\right)}{\partial \delta }\right|}_{\delta =\vec{0}}\\ S_{k}^{j}=H_{k}^{j}P_{k+1|k}^{j}{\left(H_{k}^{j}\right)}^{T}+R_{k}\\ W_{k}^{j}=P_{k+1|k}^{j}{\left(H_{k}^{j}\right)}^{T}{\left(S_{k}^{j}\right)}^{-1}\\ r_{k}^{j}=z_{k}⊟h\left({\hat{x}}_{k+1|k}^{j}\right)\\ {\hat{x}}_{k+1|k+1}^{j}={\hat{x}}_{k+1|k}^{j}⊞W_{k}^{j}r_{k}^{j}\\ J_{k}^{j}={\left.\frac{\partial \left({\hat{x}}_{k+1|k}^{j}⊞\left(r_{k}^{j}+\delta \right)⊟{\hat{x}}_{k+1|k+1}^{j}\right)}{\partial \delta }\right|}_{\delta =\vec{0}}\\ P_{k+1|k+1}^{j}=J_{k}^{j}\left(P_{k+1|k}^{j}-W_{k}^{j}S_{k}^{j}{\left(W_{k}^{j}\right)}^{T}\right){\left(J_{k}^{j}\right)}^{T}\\ \Lambda_{k}^{j}=N\left(r_{k}^{j};0,S_{k}^{j}\right)\\ \mu_{k+1|k+1}^{j}=\frac{1}{c}\Lambda_{k}^{j}c_{k}^{j},\;c\;\mathrm{is}\;a\;\mathrm{normalization}\;\mathrm{factor} \end{array}$
**Combination**	$\begin{array}{c} {\hat{x}}_{k+1|k+1}=\sum_{j=1}^{M}{\hat{x}}_{k+1|k+1}^{j}\mu_{k+1|k+1}^{j} \end{array}$ x^k+1|k+1=⊞-WeightedSumj∈[1,M](x^k|k,x^k+1|k+1j,μk+1|k+1j) $\begin{array}{c} P_{k+1|k+1}=\sum_{j=1}^{M}\mu_{k}^{j}\left(P_{k+1|k+1}^{j}+{\left[{\hat{x}}_{k+1|k+1}^{j}-{\hat{x}}_{k+1|k+1}\right]}^{⊗}\right) \end{array}$ Pk+1|k+1=⊞-WeightedCovj∈[1,M](x^k+1|k+1,Pk+1|k+1j,x^k+1|k+1j,μk+1|k+1j)

**Table 2 sensors-21-04164-t002:** The ⊞-RTSIMMS smoothing scheme based on  [11].

**Backward transition probability**	$\begin{array}{cc} b_{ij} & =\frac{1}{e_{i}}p_{ji}\mu_{k|k}^{j}\\ e_{i} & =\sum_{l=1}^{M}p_{li}\mu_{k|k}^{l} \end{array}$
**Backward mixing probability**	$\begin{array}{cc} \mu_{k+1|N}^{ij} & =\frac{1}{d_{j}}b_{ij}\mu_{k+1|N}^{i}\\ d_{j} & =\sum_{l=1}^{M}b_{li}\mu_{k+1|N}^{l} \end{array}$
**Backward mixing**	$\begin{array}{cc} {\hat{x}}_{k+1|N}^{0j} & =\underset{i\in [1,M]}{⊞\text{-}\mathrm{WeightedSum}}\left({\hat{x}}_{k+1|N}^{j},{\hat{x}}_{k+1|N}^{i},\mu_{k+1|N}^{ij}\right)\\ P_{k+1|N}^{0j} & =\underset{i\in [1,M]}{⊞\text{-}\mathrm{WeightedCov}}\left({\hat{x}}_{k+1|N}^{0j},P_{k+1|N}^{i},{\hat{x}}_{k+1|N}^{i},\mu_{k+1|N}^{ij}\right) \end{array}$
**Mode-matched smoothing**	$\begin{array}{cc} {\hat{x}}_{k|N}^{j} & ={\hat{x}}_{k|k}^{j}⊞C_{k}^{j}\left[{\hat{x}}_{k+1|N}^{0j}⊟{\hat{x}}_{k+1|k}^{j}\right]\\ P_{k|N}^{j} & =J_{k}^{j}\left(P_{k|k}^{j}+C_{k}^{j}\left[B_{k}^{j}P_{k+1|N}^{0j}B_{k}^{jT}-P_{k+1|k}^{j}\right]C_{k}^{jT}\right)J_{k}^{jT}\\ C_{k}^{j} & =P_{k|k}^{j}F_{k}^{jT}{\left(P_{k+1|k}^{j}\right)}^{-1}\\ F_{k}^{j} & ={\left.\frac{\partial \left(g_{j}\left({\hat{x}}_{k|k}^{j}⊞\delta \right)⊟{\hat{x}}_{k+1|k}^{j}\right)}{\partial \delta }\right|}_{\delta =\vec{0}}\\ B_{k}^{j} & ={\left.\frac{\partial \left(\left({\hat{x}}_{k+1|N}^{j}⊞\delta \right)⊟{\hat{x}}_{k+1|k}^{j}\right)}{\partial \delta }\right|}_{\delta =\vec{0}}\\ J_{k}^{j} & ={\left.\frac{\partial \left(\left({\hat{x}}_{k|k}^{j}⊞\left(C_{k}^{j}\left[{\hat{x}}_{k+1|N}^{0j}⊟{\hat{x}}_{k+1|k}^{j}\right]+\delta \right)\right)⊟{\hat{x}}_{k|N}^{j}\right)}{\partial \delta }\right|}_{\delta =\vec{0}} \end{array}$
**Smoothed mode probability**	$\begin{array}{cc} \Lambda_{k|N}^{j} & =\sum_{i=1}^{M}p_{ji}N\left({\hat{x}}_{k+1|N}^{i};{\hat{x}}_{k+1|k}^{j},P_{k+1|k}^{j}\right)\\ \mu_{k|N}^{j} & =\frac{1}{f}\Lambda_{k|N}^{j}\mu_{k|k}^{j},\;\;f\;\mathrm{is}\;a\;\mathrm{normalizing}\;\mathrm{factor} \end{array}$
**Combine smoothed estimate**	$\begin{array}{cc} {\hat{x}}_{k|N} & =\underset{j\in [1,M]}{⊞\text{-}\mathrm{WeightedSum}}\left({\hat{x}}_{k|k},{\hat{x}}_{k|N}^{j},\mu_{k|N}^{j}\right)\\ P_{k|N} & =\underset{j\in [1,M]}{⊞\text{-}\mathrm{WeightedCov}}\left({\hat{x}}_{k|N},P_{k|N}^{j},{\hat{x}}_{k|N}^{j},\mu_{k|N}^{j}\right) \end{array}$

## Data Availability

The Implementations and simulations are available at: https://github.com/TomLKoller/Manifold-RTS-Smoother (accessed on 16 June 2021), and https://github.com/TomLKoller/Boxplus-IMM (accessed on 16 June 2021).

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
