# Peer review of "The Interacting Multiple Model Filter and Smoother on Boxplus-Manifolds"

_sensors, 2021, doi:10.3390/s21124164_

Round 1

Reviewer 1 Report

The paper has many similarities with the one presented in [1]. I suggest to the authors depicting in detail the extension of this paper in order to focus on Smoother (Sections 5 and 6). Specifically, the lines 70-72 are exactly the same as those in [1] meaning that there is no other interest in the paper, apart from the original target of Multiple Model Filter Boxplus-Manifolds, which is the case of [1]. However, the authors have contributed to a significant extent of that work, adding novel material.

For that reason, I strongly suggest to the authors to give the appropriate attention to the manuscript and highlight the differences from [1], which are many and significant.

Reviewer 2 Report

This paper propose a linear approximation to the mixing of Gaussians and a Rauch-Tung-Striebel smoother for single models on boxplus-manifolds.

Similar to EKF, the proposed scheme uses linear approximation to improve the accuracy, but the simulation results show that it does not actually improve the accuracy, but increases a lot of computational complexity. Therefore, the practical application of this algorithm becomes relatively poor. hope the author does more theoretical analysis.

 As author said "The -update in  requires 2 additional linear approximations compared to the (M)-EKS formula,……, it seems to have the opposite effect." This indicates that the linearization degrades the data results. Does it indicate that the algorithm has not been improved properly or that the new method has not been perfected?

In this paper, a lot of formula derivation is done to extend the application of on the boxplus-method,The expression of the formula still needs to be improved, the intermediate process should be reduced, and the explanation of the key formula should be more sufficient. In particular, the change in RTS-smoother in the manifolds should be clearly indicated.

Round 2

Reviewer 2 Report

The main content of this paper is to apply the idea of extended Kalman filter to a quaternion model of BoxPlus. In general, the theoretical background is relatively strong, but the final results are not satisfactory. After the improvement, the variance of the simulation results increases. Please carefully analyze the advantages and characteristics of the algorithm, and clearly explain the existence of the original algorithm. Using Smoother to improve data accuracy may not prove the superiority of the authors' improved algorithm.
